# Metabolic Control, Diabetic Complications and Drug Therapy in a Cohort of Patients with Type 1 and Type 2 Diabetes in Secondary and Tertiary Care between 2004 and 2019

**DOI:** 10.3390/ijerph20032631

**Published:** 2023-02-01

**Authors:** Matthias Roth, Thomas Lehmann, Christof Kloos, Sebastian Schmidt, Christiane Kellner, Gunter Wolf, Nicolle Müller

**Affiliations:** 1Department of Internal Medicine III, Jena University Hospital, 07747 Jena, Germany; 2Institute of Medical Statistics, Computer and Data Sciences, Jena University Hospital, 07743 Jena, Germany

**Keywords:** diabetes mellitus, metabolic control, complications, therapy, quality of care

## Abstract

This paper studies the features of metabolic parameters, diabetic complications and drug therapy of a single-centre cohort of patients with type 1 diabetes (T1DM) or type 2 diabetes (T2DM) in secondary care and tertiary care over a 15-year period. Methods: Retrospective cross-sectional analysis of four single-centre cohorts between 2004 and 2019. All patients with T1DM or T2DM in secondary care (*n* = 5571) or tertiary care (*n* = 2001) were included. Statistical analyses were performed using linear mixed models. Results: Diabetes duration increased in both patients with T1DM and T2DM in secondary care and tertiary care (*p* < 0.001). Patients in secondary care consistently showed good glycaemic control, while patients in tertiary care showed inadequate glycaemic control. All four cross-sectional cohorts showed a significant increase in the prevalence of nephropathy over time and three out of four cohorts (T1DM and T2DM in secondary care and T2DM in tertiary care) showed an increase in the prevalence of neuropathy (all *p* < 0.001). The incidence of severe hypoglycaemia was consistently low. The use of insulin pumps and insulin analogues in the therapy of T1DM increased significantly. Conclusions: The increased prevalence of complications is likely due to older age and longer diabetes duration. Low rates of hypoglycaemia, lower limb amputations and good glycaemic control in secondary care patients indicate a good structure of patient care.

## 1. Introduction

The treatment of patients with diabetes mellitus (DM) has changed considerably over the past 15–20 years. New drugs have been approved, namely glucagon-like peptide-1 (GLP-1) receptor agonists, dipeptidyl peptidase-4 (DPP-4) inhibitors and sodium-glucose co-transporter-2 (SGLT2) inhibitors [1,2]. The results of the 2008 ACCORD study, as well as other studies, have led to changes in the recommendations of various professional societies on behalf of antihyperglycaemic therapy and glycaemic control [3]. In 2012, the European Association for the Study of Diabetes and the American Diabetes Association suggested for the first time an individualised target range, instead of fixed glycated haemoglobin (HbA1c) values [4]. In 2013, the German national health care guidelines were adapted accordingly [5]. Technological innovations also play an increasingly important role in patient care. In 2020, 67% of all patients with type 1 diabetes (T1DM) used continuous glucose monitoring (CGM). The proportion of patients with T1DM receiving insulin pump therapy (continuous subcutaneous insulin infusion; CSII) continues to increase in Germany and accounted for about 25% of the patients in 2020 [6,7]. Disease management programmes (DMP) have also improved diabetes care since their introduction in 2002 [8,9]. However, negative trends also seem to be on the rise. For example, some studies indicate an increase in the prevalence of certain diabetic complications leading to more frequent consultations [10,11]. In addition, patients with DM have 2–3 times higher health care costs than those without diabetes. In 2019, German health care expenditure for treating patients with DM accounted for approximately 9% of the total health care costs. Since 2000, treatment costs have increased by around 23.5%. The two largest cost drivers were the treatment of diabetic complications in tertiary care and drug therapy [6]. Around 80–90% of patients with DM in Germany are treated by general practitioners (primary care) and about 10–20% require a hospital outpatient clinic or specialist practice (secondary care, SC). In case of emergencies, adaptation problems and severe complications, inpatient admission is required (tertiary care, TC) [6]. Providing data on the specific care situation in Germany can help to optimize diabetes treatment and thus reduce risks and complications.

Therefore, the aim of this study is to obtain a more detailed overview regarding the time-related development of patient characteristics, complications and drug utilisation among patients with T1DM and T2DM in SC and TC to observe and analyse trends and developments over a period of 15 years between 2004 and 2019 in Germany.

## 2. Methods

### 2.1. Participants and Study Design

The present study is a retrospective cross-sectional analysis with a study period of 15 years. Four survey periods 2004, 2009, 2014 and 2019 were defined. Patients in SC and in TC were independently studied. Data were obtained from the Division of Endocrinology and Metabolic Diseases, Department of Internal Medicine III, Jena University Hospital, Germany. All patients diagnosed with T1DM or T2DM admitted either to the diabetes outpatient clinic (SC) or inpatient clinic (TC) between the first of January and December 31 of the corresponding study period were included. All analyses were performed separately according to diabetes type. Patients recorded in several survey periods could be identified by the individual patient ID but were not excluded from the analysis. In SC, 2293 (41.2%) patients were recorded in at least 2 of the survey periods; in tertiary care it was 120 (5.8%).

### 2.2. Measurements

Study participant data were obtained from digital medical records. Diabetes type and duration were recorded at the initial consultation. Blood pressure, height, weight and HbA1c were measured repeatedly at each consultation and kidney function was measured in yearly intervals in SC; all of these parameters were measured at patient admission in TC.

HbA1c was measured using high-performance liquid chromatography (TOSOH Glycohemoglobin Analyzer HLC–723 GhbV, Tosoh Corporation, Tokyo, Japan) with a normal range of 5.0–6.2%. Values were then adjusted according to the mean normal value of healthy people according to the Diabetes Control and Complications Trial [12]. Estimated glomerular filtration rate (eGFR) was calculated using the Chronic Kidney Disease Epidemiology Collaboration equation [13]. Obesity classes were categorised according to the World Health Organization (WHO) classification of classes 1–3: Class 1 obesity (30.0–34.9 kg/m^2^), Class 2 obesity (35.0–39.9 kg/m^2^) and Class 3 obesity (≥40.0 kg/m^2^) [14]. In addition, results of the WHO-5 questionnaire were assessed to evaluate the subjective well-being of patients and the presence of depression in cases of lower scores [15]. Less than 50% of the valid values were available for eGFR and WHO-5 in SC in 2004, and for WHO-5 in TC, so these variables were not evaluated. Missing data are marked n.a. (not available) in the tables.

### 2.3. Diagnosis and Therapy

Diabetic nephropathy was defined via a urine albumin-creatinine ratio ≥30 mg/g or eGFR < 60 mL/min/1.73 m^2^. Diabetic neuropathy was assessed using the Neuropathy Disability Score (NDS) and the Neurological Symptom Score (NSS). Data on retinopathy were provided by the attending ophthalmologists. Non-severe hypoglycaemia was defined by blood glucose values ≤3.9 mmol/L or the presence of characteristic symptoms (e.g., sweating, shaking, headache or impaired concentration) which improved after the intake of rapidly resorbable carbohydrates. Severe hypoglycaemia was defined if patients needed either intravenous glucose or an intramuscular glucagon injection. Foot examinations to assess neuropathy and screenings for foot lesions were performed at least in yearly intervals. Drug therapy was analysed using the Anatomical Therapeutic Chemical (ATC) classification [16]. Antihypertensive drugs include the classes of angiotensin-converting enzyme (ACE) inhibitors, beta-adrenergic receptor blockers, calcium channel blockers and sartans and thiazide diuretics, as well as fixed combinations of the individual drug classes. Antihyperlipidaemic drugs include the substance classes of statins, fibrates, anion exchange resins, cholesterol absorption inhibitors and proprotein convertase subtilisin/kexin type 9 (PCSK9) inhibitors, as well as fixed combinations of the individual drug classes. Double or triple combinations contain both fixed combinations and the sum of single preparations. Insulin therapy was classified into human insulin and insulin analogues. In addition, a subgroup analysis of T1DM patients with either CSII or intensified insulin therapy (ICT) regarding glycaemic control and hypoglycaemia was conducted for the 2019 period.

### 2.4. Statistical Analysis

Continuous data are expressed as mean ± standard deviation (SD). Categorical data are described by absolute and relative frequencies. As patients were collected in several survey periods, linear mixed models for continuous variables were fitted to account for these correlated observations. The survey period was included as a fixed effect and the patient (with their ID) was a random effect in these models. For binary variables (e.g., complications such as nephropathy or neuropathy), a generalised linear mixed model was applied to assess the difference between years. In these models, a binary link function was used with the survey period as a fixed effect and the patient as a random effect. Statistical analysis was performed using a linear mixed model for continuous variables. A generalised linear mixed model was used for categorical variables. In both cases, random effects models were used to adjust the results for repeated measures. Pairwise comparisons were made between the 4 survey periods to detect significant differences between parameters. A Bonferroni correction based on multiple testing was applied. It was also examined as to whether there was a significant trend between the survey periods. An unpaired *t*-test was performed to analyse the differences between patients with ICT and CSII for 2019. A *p*-value < 0.05 was considered statistically significant. Statistical calculation was performed using IBM SPSS Statistics version 27 (IBM Corporation, Armonk, NY, USA).

## 3. Results

In SC, 5571 individuals were included, 1192 with T1DM and 4379 with T2DM (2004: 283/933; 2009: 327/1195; 2014: 299/1143; 2019: 283/1108). In TC, 2001 individuals were included, 519 with T1DM and 1482 with T2DM (2004: 119/380; 2009: 136/365; 2014: 131/338; 2019: 133/399).

### 3.1. Characteristics and Metabolic Control

#### 3.1.1. T1DM

The mean age of patients in SC was 49.3 years (y) (Table 1) compared to 44.5 years in TC (Table 2). Age as well as diabetes duration increased significantly from 19.0 to 24.8 years in SC (*p* < 0.001) and from 17.5 to 20.2 years in TC (*p* < 0.001). HbA1c values remained stable in both care levels (SC: 7.3%, *p* = 0.118; TC: 7.7%, *p* = 0.251). Body mass index (BMI) was slightly higher in SC patients and rose in both SC (26.6 to 27.6 kg/m^2^; *p* < 0.001) and TC (25.8 to 27.2 kg/m^2^; *p* = 0.005). WHO-5 score for patients with T1DM remained unchanged. Patients in SC showed better well-being than patients in TC.

#### 3.1.2. T2DM

The mean age of patients in SC was 65.8 years compared to 69.4 years in TC. Compared to T1DM, age (SC: 63.5 to 67.5 years, *p* < 0.001; TC: 69.4 to 69.9 years, *p* < 0.001) as well as diabetes duration (SC: 12.7 to 15.5 years, *p* < 0.001; TC: 13.0 to 14.7 years, *p* < 0.001) increased significantly. In TC patients, a reduction in HbA1c from 8.1 to 7.5% was observed (*p* = 0.002), while in SC, values remained stable at about 7.0% (*p* = 0.790). In SC, BMI raised from 31.7 to 32.6 kg/m^2^ with a significantly higher rate of class three obesity (8.4 to 14.3%, *p* < 0.001). Lower BMI values were observed in TC without relevant changes among obesity classes. Regarding the WHO-5 score, there were significant improvements in well-being in patients with T2DM (SC: 13.8 to 15.8, *p* < 0.001; TC: 10.7 to 14.0, *p* < 0.001).

The eGFR decreased slightly but significantly whereas diastolic blood pressure increased in both care levels and for both types of diabetes.

### 3.2. Complications and Comorbidity

#### 3.2.1. T1DM

Complications and comorbidities of SC patients are listed in Table 3 and those of TC patients in Table 4. The prevalence of diabetic nephropathy increased markedly between 2009 and 2019 (SC: 22.6 to 37.8%, *p* < 0.001; TC: 23.5 to 42.9%, *p* = 0.001) (Figure 1 and Figure 2). A rising prevalence of diabetic neuropathy was only found in SC patients (SC: 15.2 to 27.6%, *p* < 0.001; TC: 26.1 to 27.8%, *p* = 0.060). The prevalence of diabetic retinopathy did not change significantly between 2009 and 2014 across both care levels. Foot complications occurred rarely in patients with T1DM and the prevalence remained comparable. Episodes of non-severe hypoglycaemia were more common in TC than in SC (SC: 1.82 to 1.69 events/week, *p* = 0.329; TC: 2.60 to 2.55 events/week, *p* = 0.875). The incidence of severe hypoglycaemia also remained stable (SC: 0.11 to 0.16 events/year, *p* = 0.834; TC: 0.13 to 0.29 events/year, *p* = 0.516).

About half of the patients with T1DM had hypertension with no significant changes in prevalence over time. The rate of cardiovascular complications increased significantly in secondary care (SC: 19.1 to 25.7%, *p* = 0.044; TC: 18.8 to 27.3%, *p* = 0.056).

#### 3.2.2. T2DM

Rates of diabetic nephropathy were even higher in patients with T2DM (SC: 37.9 to 56.2%, *p* < 0.001; TC: 46.6 to 67.4%, *p* < 0.001) (Figure 3 and Figure 4). The prevalence of diabetic neuropathy was higher in TC patients, showing a significant increase in both care levels (SC: 25.6 to 39.2%, *p* < 0.001; TC: 44.2 to 52.1%, *p* < 0.001). The prevalence of diabetic retinopathy was stable at around 20% in SC and only showed a decrease in prevalence in TC between 2014 and 2019 (TC: 24.2 to 15.2%, *p* < 0.05). The incidence of new foot complications was about three times higher in patients undergoing inpatient treatment but did not show any significant changes (SC: 13.0 to 10.3%, *p* = 0.126; TC: 29.1 to 29.3%, *p* = 0.963), while the rate of new lower limb amputations remained stable at about 1% in SC and 3% in TC during the study period. The rate of non-severe hypoglycaemia halved in SC and remained stable in TC patients (SC: 0.25 to 0.12 events/week, *p* < 0.001; TC: 0.37 to 0.26 events/week, *p* = 0.928). Events of severe hypoglycaemia were rare but occurred more frequently in hospitalised patients (SC: 0 to 0.02 events/year, *p* = 0.311; TC: 0.03 to 0.12 events/year, *p* = 0.060). Hypertension was common in patients with T2DM and even showed a slight increase in prevalence in SC. Overall, cardiovascular complications occurred more frequently in TC patients, who also showed a significant decrease in prevalence, while the rate in SC remained quite stable. (SC: 40.7 to 39.7%, *p* = 0.519; TC: 61.6 to 44.8%, *p* < 0.001.)

### 3.3. Antidiabetic Drug Therapy

#### 3.3.1. T1DM

Table 5 and Table 6 show the antihyperglycaemic therapy used in SC and TC. The use of human insulin in the treatment of patients with T1DM gradually declined (SC: 71.4 to 37.5%, *p* < 0.001; TC: 68.9 to 19.5%, *p* < 0.001), while the use of insulin analogues remarkably increased over the same period (SC: 23.7 to 69.6%, *p* < 0.001; TC: 17.6 to 77.4%, *p* < 0.001). CSII was used increasingly with a proportion of 30.4% in SC and 32.2% in TC in 2019. Among SC patients, the use of lipid-lowering and antihypertensive drugs increased, as did the total number of drugs (SC: 4.0 to 5.5, *p* < 0.001). In contrast, TC patients showed no significant changes in the use of lipid-lowering drugs and the total amount of drugs taken, while the utilisation of antihypertensive drugs decreased (TC: 53.8 to 38.3%, *p* = 0.029).

#### 3.3.2. T2DM

The use of antihyperglycaemic therapy (non-insulin drugs and/or insulin) decreased moderately in both SC and TC patients with T2DM (SC: 80.1 to 76.3%, *p* = 0.019; TC: 78.2 to 70.2%, *p* < 0.001). Likewise, insulin therapy became less common (SC: 64.1 to 45.3%, *p* < 0.001; TC: 63.9 to 50.4%, *p* < 0.001). Metformin was already the most commonly used non-insulin drug in 2004, but its use increased considerably (SC: 23.9 to 47.5%, *p* < 0.001; TC: 11.6 to 31.3%, *p* < 0.001). Sulfonylureas were the second-most-used drugs after metformin in 2004 and held their position in SC but not in TC (SC: 7.6 to 7.5%, *p* = 0.468; TC: 10.0 to 3.0%, *p* < 0.001). After the introduction to the market of DPP-4 inhibitors, GLP-1 receptor agonists and SGLT-2 inhibitors found their way into the treatment of T2DM. In 2019, the shares of utilisation were 5.4%, 6.4% and 11.1% for SC and 6.8%, 6.8% and 6.5% for TC, respectively. Dual combinations were used more frequently (SC: 3.3 to 16.7%, *p* < 0.001; TC: 2.1 to 12.0%, *p* < 0.001). The use of lipid-lowering drugs increased between 2004 and 2019 (SC: 40.5 to 49.5%, *p* < 0.001; TC: 20.8 to 41.9%, *p* < 0.001). About 85% of the patients in SC were treated with antihypertensive agents over the entire period, while their prevalence increased in TC (TC: 61.9 to 88.2%, *p* < 0.001).

The total number of utilised medications increased in both care levels (SC: 6.1 to 7.9, *p* < 0.001; TC: 5.6 to 9.2, *p* < 0.001).

### 3.4. Comparison of T1DM Patients Treated with Either ICT or CSII

The proportion of patients receiving ICT decreased in favour of CSII therapy (SC: 72.4 to 67.5%; TC: 78.7 to 66.7%) (Table 7 and Table 8). HbA1c was better among patients in SC using CSII (SC: −0.3%, *p* = 0.037; TC: −0.4%, *p* = 0.399). While there was no significant difference between the two groups regarding severe hypoglycaemia (SC: 0.17 vs. 0.09 events/year, *p* = 0.216; TC: 0.44 vs. 0.15 events/year, *p* = 0.133), the incidence of non-severe hypoglycaemia was higher in SC patients with CSII therapy (SC: 2.22 vs. 1.32 events/week, *p* = 0.002; TC: 2.59 vs. 2.12 events/week, *p* = 0.507).

## 4. Discussion

### 4.1. Characteristics and Metabolic Control

Diabetes duration increased across the cohort, in both patients with T1DM and T2DM and in both outpatient and inpatient care. This increase could be explained by the generally increasing life expectancy of diabetic patients and the overall earlier onset and diagnosis of the disease, especially T2DM [17,18]. In addition, several diabetic specialist practices started to operate in the service area of the university hospital during this time, which drained some of the younger patients treated in the hospital. The trend towards increasing BMI values is not limited to Germany but can also be observed in other Western countries [19]. The outpatients showed good glycaemic control and were consistently within the target range, while the inpatients showed inadequate glycaemic control. This negative selection can be explained by the staged treatment system for patients with DM in Germany, consisting of primary, secondary and tertiary care levels. The higher the level of care, the higher the HbA1c values and the prevalence of secondary complications [20,21]. However, our study showed a clinically relevant improvement in HbA1c of 0.6% in the inpatients with T2DM. Long-term data from the DMP North Rhine region in Germany also demonstrated an improvement in HbA1c for patients with T2DM between 2011 and 2020. The rate of patients within the target range improved from 58% to 62% [22]. While older national guidelines still recommend a rigid HbA1c target of ≤6.5% for the treatment of T2DM, current guidelines recommend a patient-oriented, individual target range between 6.5% and 7.5%, with a target range of up to <8.5% for people with limited life expectancy or advanced comorbidities [23,24,25]. In adults with T1DM, according to the German guideline, the HbA1c target ranges between 6.5% and 7.5% [24].

Besides stable weight and glycaemic control, blood pressure control also plays a key role in the treatment of DM. The association between diabetes, hypertension and increased risk of cardiovascular complications has been known for a long time [26]. A large proportion of the patients studied was within the target range for optimal blood pressure control. Surprisingly, TC patients had better blood pressure control compared to SC patients. Only the outpatients with T2DM missed the mean target range. Compared to long-term data from the DMP North Rhine, around 58% of patients with T2DM consistently achieved a target value of <140/90 mmHg between 2001 and 2020. The rate of patients with T1DM within the blood pressure target decreased from 60% in 2011 to 55% in 2020 [22]. The recommendation for optimal blood pressure control also changed during the last few years. While older guidelines recommend values <130/85 mmHg for patients with T2DM, current health care guidelines suggest 140/90 mmHg, with the possibility of individual adaptation [23,25].

Since individuals with DM are prone to psychological comorbidities, and psychological impairment affects metabolic control, it is important to assess mental health [27]. The WHO-5 questionnaire is an easily used instrument to screen for subjective well-being and depression [15]. Patients in TC had lower scores than those in SC, probably due to a greater burden of disease. The significant improvement in the WHO-5 score in patients with DM2 could be explained by the declining use of insulin during the observation period, as reported in other studies [28].

### 4.2. Complications and Comorbidity

As expected, the prevalence of all diabetic complications studied was higher in TC patients than in SC patients. This finding is consistent with the recommendation to hospitalise patients with chronic hyperglycaemia, recurrent hypoglycaemia, acute diabetic foot complications or the presence of specific diabetic complications [21]. The prevalence of nephropathy was higher in patients with T2DM and increased significantly in all four patient cohorts. Other studies showed varying rates, but a prevalence of up to 45% was also reported [29,30]. Increasing age and diabetes duration probably mostly contributed to the increase in nephropathy and other complications. However, when interpreting our data, it should be noted that the prevalence of nephropathy may have been overestimated due to the retrospective design. The diagnosis was based only on single pathological laboratory results and not on two examinations within three months, as stated in the recommendation for the diagnosis of diabetic nephropathy [31]. In addition, renal function may often be worse on admission than at discharge after recompensation, rehydration and discontinuation of nephrotoxic drugs. Albuminuria may also be falsely present due to infection, extreme hyperglycaemia or other primary renal diseases. Finally, the cause of nephropathy could have had a differing aetiology from diabetes. There were no significant changes in the prevalence of retinopathy. The fact that patients with T1DM were more frequently affected by retinopathy compared to patients with T2DM was consistent with other national data from Germany [32,33].

Cardiovascular complications occurred more frequently in patients with T2DM. However, outpatients with T1DM showed a significant increase in prevalence, which was probably also related to rising age and diabetes duration. In contrast, a significant decrease in prevalence among inpatients with T2DM was found, eventually related to the improved management of risk factors, e.g., blood pressure and statins [34]. Cardiovascular complications could also have been promoted by conditions other than diabetes, e.g., cardiac arrhythmias, coagulation disorders or genetic factors.

The prevalence of diabetic neuropathy increased significantly and was higher in patients with T2DM. Depending on diagnostic criteria, examination methods and the population examined, the literature data report a prevalence of up to 31% [29,35,36]. As with other diabetic complications, another aetiology may be present, e.g., alcohol-induced polyneuropathy, a generalised neurological or multisystem disorder such as amyloid polyneuropathy or porphyria or so-called treatment-induced neuropathy. This form of neuropathy can occur when rapid recovery of glycaemic control is achieved after previous poor glycaemic control [37]. However, there are clues that screening for diabetic neuropathy in general may be carried out too irregularly in Germany, resulting in low diagnosis rates [38]. Therefore, our higher prevalence could also indicate high screening rates. Foot examinations at regular intervals are important because neuropathy can lead to serious foot complications or even amputation [35,39].

Diabetic foot complications are still the most common cause of non-traumatic limb amputations in Germany and still are often performed too early [40,41]. Data from Germany show widely varying amputation rates. Whereas rates in specialised foot centres are low at around 3%, high rates of up to 10% were recorded in regular care hospitals [6]. Incidences of almost 30% for inpatients with T2DM may be explained by the fact that the hospital studied is a certified centre of foot care. Furthermore, consistently low amputation rates in the range of 2.5–4.1% were demonstrated for patients with T2DM.

Hypoglycaemia can occur in patients with both T1DM and T2DM, but is significantly more common in patients with T1DM, which is also reflected in our study [42]. Amongst antihyperglycaemic drugs, insulin and sulfonylureas have the highest potential to cause hypoglycaemia [25]. These potentially life-threatening emergencies are often limiting to reaching optimal glycaemic control [43]. Our results showed that outpatients with T1DM had a consistently low incidence of severe hypoglycaemia, whereas severe hypoglycaemia was documented more frequently in inpatients. The decrease in non-severe hypoglycaemia in T2DM outpatients may be explained by less of a use of insulin therapy or less stringent target ranges for glycaemic control. Additionally, teaching programmes help to prevent hypoglycaemia [24,43,44]. Interestingly, the frequency of hypoglycaemia in patients with T2DM did not change significantly in TC, although insulin and sulfonylureas were used less frequently during the study period.

### 4.3. Antidiabetic Drug Therapy

The use of CSII increased during the study period, while the use of ICT decreased. About 30% of patients with T1DM had CSII therapy in 2019, while about 67% had ICT. The increasing use of CSII therapy is also reported in other studies from Germany and there are probably several reasons for this. First, CSII therapy is associated with better blood glucose control and less events of severe hypoglycaemia, although events of non-severe hypoglycaemia are not reduced [7,45,46]. Another factor for the increasing number of CSII may be the relatively easy approval of technical devices by German health insurance. Our results also showed better glycaemic control for patients with CSII. No significant difference in severe hypoglycaemia was demonstrated for the 2019 study period, although there was a visible trend in favour of CSII. Interestingly, patients with CSII in SC had a higher incidence of non-severe hypoglycaemia. The proportion of patients treated with insulin analogues has increased steadily since 2004, while human insulins were used less frequently. This trend can be observed nationwide when analysing the totality of all insulin prescriptions [47]. So far, no endpoint studies have shown the superiority of either insulin [48]. Our results also showed no significant changes in glycaemic control or the occurrence of hypoglycaemia. Apart from metabolic control, insulin analogues are often more convenient to use, e.g., with formulations with higher concentrated insulin, or products with ultralong effects which contribute to the increasing market share. The reduced use of insulin in patients with T2DM may be due to the fact that there is a wider choice of therapies which do not need to be administered by injection due to the approval of newer antidiabetic drugs. A significant decrease in antihyperglycaemic therapy was observed in patients with T2DM, most likely due to a significant decrease in the use of insulin as part of drug therapy. Studies demonstrated that GLP-1 receptor agonists can delay the onset of insulin therapy without the typical side effects of hypoglycaemia or weight gain and even better blood pressure control [49]. Metformin remained the most commonly prescribed oral antidiabetic drug, which was consistent with data from Germany and Europe, and is still the first-line therapy in the current treatment algorithm [1,25,47]. A reduction in the use of sulfonylureas was only observed in TC, while utilisation rates remained stable in SC. The large increase in dual combination therapy may also be related to the more widespread use of SGLT-2 inhibitors and GLP-1 receptor agonists, as these agents demonstrated benefits on cardiovascular and renal endpoints in patients with T2DM [50,51]. The current guideline recommends, after individual consideration, the initial combination of metformin with SGLT2 inhibitors or GLP-1 receptor agonists as the first-line therapy in patients with cardiovascular disease or clinically relevant renal disease [25]. Patients with T2DM in TC showed a significant increase in polypharmacy with only a small increase in age. This also indicates an increasing proportion of multimorbid patients with rising blood glucose control problems.

A considerable limitation of this study is its retrospective design, which may be prone to error. Examination results or prescriptions that were documented incorrectly or not at all could not be analysed. Some parameters, e.g., non-severe hypoglycaemia, had not yet been recorded in 2004. Only monocentric data with a characteristic profile of care were evaluated; therefore, data from several facilities may lead to different results. Some of the patients received medical treatment in more than one survey period and were therefore assessed more than once. This affected 2293 patients (41.2%) in SC and 120 patients (5.8%) in TC. However, the linear mixed models allowed for statistical analysis of significant differences without multiple patients changing the significance value. Multiple consultations nevertheless led to small biases in the results of the evaluation. Furthermore, diabetes specialists established new practices which may have contributed to a shift in the cohort with more severely ill patients in our cohort. We point out once again that a retrospective evaluation of digital routine data has a potential for error and the data must therefore be interpreted with caution. The high prevalence of diabetic complications could be an overestimation due to other aetiologies, as described in the discussion section. The strength of this study is the large number of included patients with T1DM and T2DM and the consideration of two levels of care, as well as the long observation period. The continuous annual recording of well-being by means of WHO-5 is also a special feature.

## 5. Conclusions

HbA1c levels remained stable and within the target range in both patients with T1DM and T2DM in SC. The increase in the prevalence of nephropathy and neuropathy was probably related to the increase in age and duration of diabetes. The patterns of antihyperglycaemic drug therapy were mostly consistent with other studies from Germany. The proportion of patients with CSII therapy increased sharply. Our results also demonstrate the described increase in insulin analogues in favour of human insulin. In the care of patients with T2DM, there was a decrease in the use of insulin with a simultaneous increase in the use of new antidiabetic drugs. The established drug metformin remained the most commonly used oral antidiabetic drug in the therapy of T2DM. Due to negative selection and in line with current recommendations in the national guideline and German DMP, TC patients mainly had inadequate glycaemic control, more frequent episodes of severe hypoglycaemia and a higher prevalence of diabetic complications. Compared to standard care hospitals in Germany, amputation rates were consistently low. SC patients with CSII had better glycaemic control than patients with ICT, while the incidence of non-severe hypoglycaemia was higher.

## Figures and Tables

**Figure 1 ijerph-20-02631-f001:**
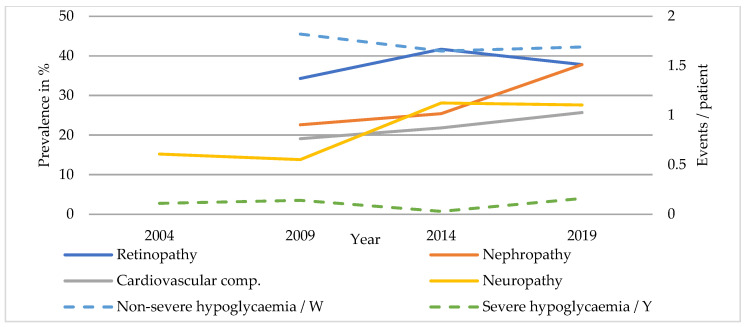
Diabetic complications of patients with T1DM in SC.

**Figure 2 ijerph-20-02631-f002:**
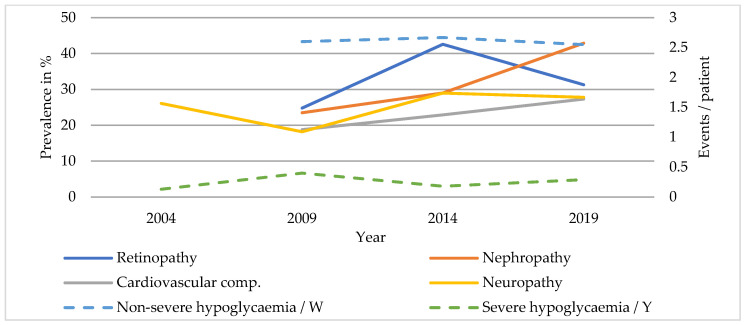
Diabetic complications of patients with T1DM in TC.

**Figure 3 ijerph-20-02631-f003:**
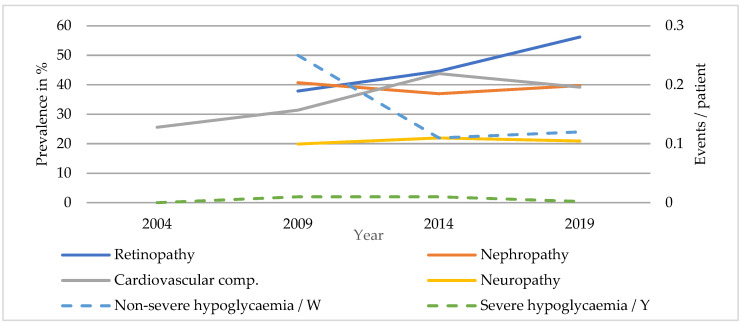
Diabetic complications of patients with T2DM in SC.

**Figure 4 ijerph-20-02631-f004:**
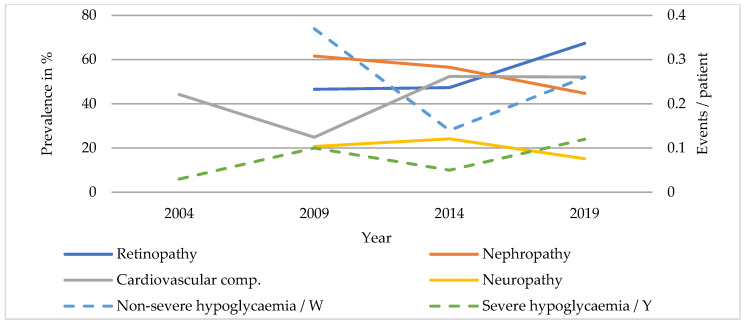
Diabetic complications of patients with T2DM in TC.

**Table 1 ijerph-20-02631-t001:** Characteristics and metabolic control of patients in secondary care (SC).

Type 1 Diabetes (T1DM)	2004	2009	2014	2019	Total	*p*-Value for Trend
Number of data sets *n* (%)	283 (23.7)	327 (27.4)	299 (25.1)	283 (23.7)	1192	
Female *n* (%)	151 (53.4)	182 (55.7)	162 (54.2)	147 (51.9)	642 (53.9)	
Age (years)	45.7 (14.8)	47.3 (16.2) *	51.4 (16.9) *	53.3 (17.6) *	49.3 (16.6)	<0.001
Diabetes duration (years)	19.0 (12.4)	19.7 (12.9) *	22.4 (13.5) *	24.8 (14.3) *	21.4 (13.4)	<0.001
BMI (kg/m^2^)	26.6 (3.9)	26.8 (4.7) *	27.3 (5.0) *	27.6 (5.1)	27.1 (4.7)	<0.001
HbA1c (%)	7.3 (1.1)	7.5 (1.0) *	7.1 (1.1) *	7.4 (1.1) *	7.3 (1.1)	0.118
BP systolic (mmHg)	138.0 (18.3)	138.6 (19.2)	136.2 (19.2)	139.3 (19.2) *	138.0 (19.0)	0.312
BP diastolic (mmHg)	79.0 (9.8)	83.3 (11.6) *	79.9 (11.89) *	81.2 (12.1)	80.9 (11.5)	0.619
eGFR (mL/min/1.73 m^2^)	n.a.	83.1 (23.5)	87.0 (25.7)	81.0 (26.2) *	83.9 (25.3)	<0.001
WHO-5 (range 0–25)	n.a.	14.9 (5.1)	15.4 (5.2)	15.9 (4.5)	15.5 (4.9)	0.150
**Type 2 Diabetes (T2DM)**						
Number of data sets *n* (%)	933 (21.3)	1195 (27.3)	1143 (26.1)	1108 (25.3)	4379	
Female *n* (%)	435 (46.6)	554 (46.4)	511 (44.7)	491 (44.3)	1991 (45.5)	
Age (years)	63.5 (11.6)	65.2 (11.8) *	66.9 (11.8) *	67.5 (12.0) *	65.8 (11.99	<0.001
Diabetes duration (years)	12.7 (9.0)	13.4 (9.4) *	15.6 (10.3) *	15.5 (10.6) *	14.3 (9.9)	<0.001
BMI (kg/m^2^)	31.7 (5.9)	32.3 (6.3) *	32.7 (6.8)	32.6 (7.4)	32.4 (6.6)	0.002
WHO class 1 obesity *n* (%)	310 (34.1)	375 (32.5)	318 (30.2)	306 (30.3)	1309 (31.7)	0.071
WHO class 2 obesity *n* (%)	139 (15.3)	202 (17.5)	181 (17.2)	153 (15.1)	675 (16.4)	0.799
WHO class 3 obesity *n* (%)	76 (8.4)	123 (10.6)	141 (13.4)	145 (14.3)	485 (11.8)	<0.001
HbA1c (%)	7.1 (1.1)	7.1 (1.0)	6.9 (1.1) *	7.0 (1.1) *	7.0 (1.1)	0.790
BP systolic (mmHg)	148.0 (22.5)	145.2 (21.3) *	140.4 (19.5) *	144.8 (20.2) *	144.5 (21.0)	<0.001
BP diastolic (mmHg)	79.5 (12.3)	84.8 (12.8) *	79.6 (12.9) *	82.9 (12.9) *	81.8 (13.0)	0.029
eGFR (mL/min/1.73 m^2^)	n.a.	69.5 (21.0)	69.8 (23.3) *	67.3 (23.4) *	68.8 (22.8)	<0.001
WHO-5 (range 0–25)	n.a.	13.8 (6.1)	15.5 (5.5) *	15.8 (5.1)	15.3 (5.5)	<0.001

Abbreviations: HbA1c = glycated haemoglobin; BMI = body mass index; BP = blood pressure; eGFR = estimated glomerular filtration rate; WHO = World Health Organization; n.a. = not available. The standard deviation for age, diabetes duration, BMI, HbA1c, blood pressure, eGFR and WHO-5 is given in brackets. * = *p* < 0.05 compared to previous survey period.

**Table 2 ijerph-20-02631-t002:** Characteristics and metabolic control of patients in tertiary care (TC).

Type 1 Diabetes (T1DM)	2004	2009	2014	2019	Total	*p*-Value for Trend
Number of data sets *n* (%)	119 (22.9)	136 (26.2)	131 (25.2)	133 (25.6)	519	
Female *n* (%)	69 (58.5)	76 (55.9)	75 (57.3)	73 (54.9)	293 (56.6)	
Age (years)	45.5 (18.9)	41.3 (15.5) *	44.3 (16.9) *	47.2 (19.7) *	44.5 (17.8)	<0.001
Diabetes duration (years)	17.5 (14.1)	16.0 (13.1) *	19.5 (12.1) *	20.2 (12.8) *	18.2 (13.1)	<0.001
BMI (kg/m^2^)	25.8 (3.7)	25.9 (4.6)	27.0 (5.4)	27.2 (6.1)	26.5 (5.1)	0.005
HbA1c (%)	7.9 (1.8)	7.9 (1.8)	7.3 (1.6) *	7.8 (1.8)	7.7 (1.7)	0.251
BP systolic (mmHg)	133.8 (19.0)	132.2 (19.1)	136.0 (19.9)	131.3 (16.6)	133.3 (18.8)	0.682
BP diastolic (mmHg)	76.0 (11.7)	80.2 (11.2) *	82.0 (11.4)	79.1 (12.0)	79.4 (11.7)	0.025
eGFR (mL/min/1.73 m^2^)	90.0 (30.1)	88.5 (22.4)	89.7 (30.0)	83.3 (30.9) *	87.6 (28.5)	0.005
WHO-5 (range 0–25)	n.a.	12.8 (6.5)	10.8 (6.2)	14.0 (6.1)	13.4 (6.2)	0.299
**Type 2 Diabetes (T2DM)**						
Number of data sets *n* (%)	380 (25.6)	365 (24.6)	338 (22.8)	399 (26.9)	1482	
Female *n* (%)	214 (56.6)	202 (55.3)	174 (51.5)	170 (42.6)	760 (51.4)	
Age (years)	69.4 (13.8)	67.9 (14.2) *	70.6 (14.0) *	69.9 (13.9) *	69.4 (14.0)	<0.001
Diabetes duration (years)	13.0 (10.1)	14.0 (10.2)	14.5 (11.6) *	14.7 (11.7) *	14.1 (10.9)	<0.001
BMI (kg/m^2^)	30.6 (6.5)	33.1 (7.5) *	32.1 (7.1)	31.7 (7.3)	31.8 (7.1)	0.018
WHO class 1 obesity *n* (%)	81 (26.0)	82 (28.9)	83 (27.9)	90 (27.9)	336 (27.6)	0.666
WHO class 2 obesity *n* (%)	39 (12.5)	50 (17.6)	53 (17.8)	51 (15.8)	193 (15.9)	0.279
WHO class 3 obesity *n* (%)	30 (9.6)	44 (15.5)	34 (11.4)	34 (10.5)	142 (11.7)	0.911
HbA1c (%)	8.1 (2.0)	7.9 (1.7)	7.6 (1.8)	7.5 (1.9)	7.6 (1.9)	0.002
BP systolic (mmHg)	140.6 (23.3)	138.3 (24.1)	142.7 (22.7)	136.4 (21.7) *	139.5 (23.1)	0.139
BP diastolic (mmHg)	74.0 (14.1)	77.0 (15.4) *	77.7 (13.2)	78.4 (13.7)	76.8 (14.2)	<0.001
eGFR (mL/min/1.73 m^2^)	65.7 (27.4)	63.9 (24.2)	62.7 (29.5)	60.8 (26.4)	62.2 (27.1)	0.017
WHO-5 (range 0–25)	n.a.	10.7 (6.4)	13.4 (6.0)	14.0 (6.2)	13.5 (6.3)	<0.001

* = *p* < 0.05 compared to previous survey period.

**Table 3 ijerph-20-02631-t003:** Diabetic complications and comorbidity of patients in secondary care (SC).

Type 1 Diabetes (T1DM)	2004	2009	2014	2019	Total	*p*-Value for Trend
Nephropathy *n* (%)	n.a.	74 (22.6)	76 (25.4)	107 (37.8) *	278 (23.3)	<0.001
Neuropathy *n* (%)	43 (15.2)	45 (13.8)	84 (28.1) *	78 (27.6)	250 (21.0)	<0.001
Retinopathy *n* (%)	n.a.	81 (34.3)	90 (41.7)	84 (37.8)	255 (37.8)	0.156
Non-severe hypoglycaemia (events/week)	n.a.	1.82 (2.11)	1.65 (2.06)	1.69 (2.63)	1.71 (2.28)	0.329
Severe hypoglycaemia (events/year)	0.11 (0.51)	0.14 (0.63)	0.03 (0.20)	0.16 (0.63)	0.11 (0.53)	0.834
New foot-related complications *n* (%)	n.a.	10 (4.8)	7 (2.9)	7 (3.4)	25 (3.3)	0.645
New amputations *n* (%)	0	1 (0.3)	0	0	1 (0.1)	0.969
Hypertension *n* (%)	n.a.	129 (54.7)	118 (54.6)	126 (56.8)	373 (55.3)	0.364
Cardiovascular complications *n* (%)	n.a.	45 (19.1)	47 (21.8)	57 (25.7)	149 (22.1)	0.044
**Type 2 Diabetes (T2DM)**						
Nephropathy *n* (%)	n.a.	453 (37.9)	510 (44.6) *	623 (56.2) *	1668 (38.1)	<0.001
Neuropathy *n* (%)	239 (25.6)	375 (31.4) *	501 (43.8) *	434 (39.2)	1549 (35.4)	<0.001
Retinopathy *n* (%)	n.a.	187 (19.9)	192 (22.0)	183 (20.9)	562 (20.9)	0.146
Non-severe hypoglycaemia	n.a.	0.25 (0.66)	0.11 (0.49)	0.12 (0.66)	0.12 (0.60)	<0.001
Severe hypoglycaemia	0.00 (0.07)	0.01 (0.15)	0.01 (0.10)	0.02 (0.29)	0.01 (0.18)	0.311
New foot-related complications *n* (%)	n.a.	110 (13.0)	83 (9.3)	79 (10.3)	288 (10.3)	0.126
New amputations *n* (%)	8 (0.9)	13 (1.1)	13 (1.1)	4 (0.4)	38 (0.9)	0.733
Hypertension *n* (%)	n.a.	786 (83.4)	758 (86.9) *	768 (87.6)	2312 (85.9)	0.006
Cardiovascular complications *n* (%)	n.a.	383 (40.7)	323 (37.0)	348 (39.7)	1054 (39.2)	0.519

Abbreviations: pat. = patient; n.a. = not available. The standard deviation for non-severe and severe hypoglycaemia is given in brackets. * = *p* < 0.05 compared to previous survey period.

**Table 4 ijerph-20-02631-t004:** Diabetic complications and comorbidity of patients in tertiary care (TC).

Type 1 Diabetes (T1DM)	2004	2009	2014	2019	Total	*p*-Value
Nephropathy *n* (%)	n.a.	32 (23.5)	38 (29.0)	57 (42.9) *	140 (27.0)	0.001
Neuropathy *n* (%)	31 (26.1)	30 (18.2) *	38 (29.0) *	37 (27.8)	113 (21.8)	0.060
Retinopathy *n* (%)	n.a.	41 (24.8)	80 (42.6) *	50 (31.3)	171 (33.3)	0.130
Non-severe hypoglycaemia (events/week)	n.a.	2.6 (2.83)	2.67 (2.78)	2.55 (4.04)	2.60 (3.31)	0.875
Severe hypoglycaemia (events/year)	0.13 (0.50)	0.40 (2.77)	0.18 (0.64)	0.29 (0.92)	0.25 (1.55)	0.516
New foot-related complications *n* (%)	n.a.	3 (4.3)	8 (7.2)	5 (4.7)	16 (5.6)	0.987
New amputations *n* (%)	0	1 (0.7)	1 (0.8)	0	2 (0.4)	0.993
Hypertension *n* (%)	n.a.	82 (49.7)	101 (53.7)	81 (50.3)	264 (51.4)	0.653
Cardiovascular complications *n* (%)	n.a.	31 (18.8)	43 (22.9)	44 (27.3)	118 (23.0)	0.056
**Type 2 Diabetes (T2DM)**						
Nephropathy *n* (%)	n.a.	170 (46.6)	160 (47.3)	269 (67.4) *	624 (42.1)	<0.001
Neuropathy *n* (%)	168 (44.2)	91 (24.9) *	177 (52.4) *	208 (52.1)	644 (43.5)	<0.001
Retinopathy *n* (%)	n.a.	89 (20.7)	99 (24.2)	56 (15.2) *	244 (20.2)	0.162
Non-severe hypoglycaemia (events/week)	n.a.	0.37 (1.20)	0.14 (0.61)	0.26 (0.84)	0.23 (0.85)	0.928
Severe hypoglycaemia (events/year)	0.03 (0.17)	0.10 (0.42)	0.05 (0.28)	0.12 (0.62)	0.08 (0.43)	0.060
New foot-related complications *n* (%)	n.a.	44 (29.1)	77 (28.8)	86 (29.3)	207 (29.1)	0.963
New amputations *n* (%)	11 (2.9)	11 (3.0)	14 (4.1)	10 (2.5)	46 (3.1)	0.974
Hypertension *n* (%)	n.a.	364 (84.7)	338 (82.6)	312 (84.8)	1014 (84.0)	0.954
Cardiovascular complications *n* (%)	n.a.	265 (61.6)	231 (56.5)	165 (44.8) *	661 (54.8)	<0.001

* = *p* < 0.05 compared to previous survey period.

**Table 5 ijerph-20-02631-t005:** Drug therapy of patients in secondary care.

Type 1 Diabetes (T1DM)	2004	2009	2014	2019	Total	*p*-Value
Human insulin *n* (%)	202 (71.4)	173 (52.9) *	134 (44.8)	106 (37.5)	615 (51.6)	<0.001
Insulin analogues *n* (%)	67 (23.7)	145 (44.3) *	170 (56.9) *	197 (69.6) *	579 (48.6)	<0.001
CSII *n* (%)	69 (25.4)	75 (25.1)	70 (25.1)	72 (30.4)	286 (26.3)	<0.001
Antihypertensive drugs *n* (%)	125 (44.2)	166 (50.8)	171 (57.2)	166 (58.7)	628 (52.7)	<0.001
Antihyperlipidaemic drugs *n* (%)	38 (13.4)	74 (22.6) *	86 (28.8)	81 (28.6)	279 (23.4)	<0.001
Total number of medications (*n*)^2^	4.0 (3.7–4.3)	4.8 (4.4–5.2) *	5.2 (4.7–5.6) *	5.5 (5.0–6.0) *	4.9 (4.6–5.1)	<0.001
**Type 2 Diabetes (T2DM)**						
Antidiabetic drugs *n* (%)	747 (80.1)	991 (82.9)	891 (78.0) *	845 (76.3)	3474 (79.3)	0.019
Insulin *n* (%)	598 (64.1)	705 (59.0)	625 (54.7)	502 (45.3) *	2430 (55.5)	<0.001
Metformin *n* (%)	223 (23.9)	477 (39.9) *	477 (41.7)	526 (47.5) *	1703 (38.9)	<0.001
Sulfonylureas *n* (%)	71 (7.6)	96 (8.0)	87 (7.6)	83 (7.5)	337 (7.7)	0.468
DPP-4 inhibitors *n* (%)	0	27 (2.3)	70 (6.1) *	60 (5.4)	157 (3.6)	<0.001
SGLT-2 inhibitors *n* (%)	0	0	13 (1.1)	123 (11.1) *	136 (3.1)	<0.001
GLP-1 receptor agonists *n* (%)	0	11 (0.9)	13 (1.1)	71 (6.4) *	95 (2.2)	<0.001
Dual therapy *n* (%)	31 (3.3)	77 (6.4) *	121 (10.6) *	185 (16.7) *	414 (9.5)	<0.001
Triple therapy *n* (%)	0	7 (0.6)	23 (2.0)	18 (1.6)	48 (1.1)	0.17
Antihypertensive drugs *n* (%)	774 (83.0)	1050 (87.9) *	951 (83.2) *	912 (82.3)	3687 (84.2)	0.357
Antihyperlipidaemic drugs *n* (%)	378 (40.5)	605 (50.6) *	571 (50.0)	548 (49.5)	2102 (48.0)	<0.001
Total number of medications (*n*)^2^	6.1 (5.9–6.3)	7.0 (6.8–7.2) *	7.4 (7.2–7.6) *	7.9 (7.7–8.2) *	7.1 (7.0–7.2)	<0.001

Abbreviations: CSII = continuous subcutaneous insulin infusion; DPP-4 = dipeptidyl peptidase-4; SGLT-2 = sodium–glucose co-transporter 2; GLP-1 = glucagon-like peptide 1. ^2^ Mean value (95% confidence interval). * = *p* < 0.05 compared to previous survey period.

**Table 6 ijerph-20-02631-t006:** Drug therapy of patients in tertiary care (TC).

Type 1 Diabetes (T1DM)	2004	2009	2014	2019	Total	*p*-Value for Trend
Human insulin *n* (%)	82 (68.9)	57 (41.9) *	22 (16.8) *	26 (19.5)	187 (36.0)	<0.001
Insulin analogues *n* (%)	21 (17.6)	76 (55.9) *	101 (77.1) *	103 (77.4)	301 (58.0)	<0.001
CSII *n* (%)	17 (15.7)	39 (33.1) *	37 (30.3)	28 (32.2)	121 (27.8)	0.023
Antihypertensive drugs *n* (%)	64 (53.8)	39 (28.7) *	37 (28.2)	51 (38.3)	191 (36.8)	0.029
Antihyperlipidaemic drugs *n* (%)	22 (18.5)	11 (8.1)	20 (15.3)	22 (16.5)	75 (14.5)	0.839
Total number of medications (*n*)	5.1 (4.5–5.8)	3.6 (2.9–4.2) *	3.6 (2.9–4.3)	5.0 (4.1–5.9) *	4.3 (4.0–4.7)	0.751
**Type 2 Diabetes (T2DM)**						
Antidiabetic drugs *n* (%)	297 (78.2)	281 (77.0)	199 (58.9) *	280 (70.2) *	1057 (71.3)	<0.001
Insulin *n* (%)	243 (63.9)	203 (55.6)	158 (46.7)	201 (50.4)	805 (54.3)	<0.001
Metformin *n* (%)	44 (11.6)	86 (23.6) *	62 (18.3)	125 (31.3) *	317 (21.4)	<0.001
Sulfonylureas *n* (%)	38 (10.0)	41 (11.2)	4 (1.2) *	12 (3.0)	95 (6.4)	<0.001
DPP-4 inhibitors *n* (%)	0	5 (1.4)	26 (7.7) *	27 (6.8)	58 (3.9)	<0.001
SGLT-2 inhibitors *n* (%)	0	0	4 (1.2)	26 (6.5)	30 (2.0)	0.005
GLP-1 receptor agonists *n* (%)	0	9 (2.5)	7 (2.1)	27 (6.8)	43 (2.9)	0.017
Dual therapy *n* (%)	8 (2.1)	22 (6.0)	27 (8.0)	48 (12.0)	105 (7.1)	<0.001
Triple therapy *n* (%)	0	2 (0.5)	6 (1.8)	5 (1.3)	13 (0.9)	0.517
Antihypertensive drugs *n* (%)	221 (61.9)	261 (75.7) *	198 (78.9) *	298 (88.2) *	978 (75.8)	<0.001
Antihyperlipidaemic drugs *n* (%)	79 (20.8)	137 (37.5) *	111 (32.8)	167 (41.9) *	494 (33.3)	<0.001
Total number of medications (*n*)	5.6 (5.2–6.1)	7.2 (6.7–7.6) *	7.4 (6.8–7.9)	9.2 (8.7–9.7) *	7.3 (7.1–7.6)	<0.001

* = *p* < 0.05 compared to previous survey period.

**Table 7 ijerph-20-02631-t007:** Comparison between T1DM patients treated with ICT or CSII in secondary care (SC).

Number of T1DM Patients *n* (%)	2004	2009	2014	2019	Total	*p* for Trend	*p* between ICT and CSII 2019
ICT	197 (72.4)	216 (72.2)	200 (71.7)	160 (67.5)	773 (71.1)	0.008	
CSII	69 (25.4)	75 (25.1)	70 (25.1)	72 (30.4)	286 (26.3)	<0.001	
HbA1c (%)							0.037
ICT	7.3 (1.1)	7.5 (1.1) *	7.2 (1.1) *	7.5 (1.1)	7.4 (1.1)	0.558	
CSII	7.2 (0.9)	7.5 (0.9) *	6.8 (1.1) *	7.2 (0.9) *	7.2 (1.0)	0.225	
Non-severe hypoglycaemia (events/week)							0.002
ICT	n.a.	1.75 (2.22)	1.52 (1.90)	1.32 (2.00)	1.53 (2.02)	0.047	
CSII	n.a.	2.25 (1.78)	2.18 (2.43)	2.22 (2.21)	2.20 (2.19)	0.942	
Severe hypoglycaemia (events/year)							0.216
ICT	0.12 (0.58)	0.14 (0.62)	0.04 (0.19)	0.17 0.67)	0.12 (0.55)	0.895	
CSII	0.07 (0.25)	0.18 (0.68)	0	0.09 (0.37)	0.08 (0.41)	0.627	

Abbreviations: ICT = intensified insulin therapy; CSII = continuous subcutaneous insulin infusion; pat. = patient; n.a. = not available. The standard deviation for HbA1c and both non-severe and severe hypoglycaemia is given in brackets. * = *p* < 0.05 compared to previous survey period.

**Table 8 ijerph-20-02631-t008:** Comparison between T1DM patients treated with ICT or CSII in tertiary care.

Number of T1DM Patients *n* (%)	2004	2009	2014	2019	Total	*p* for Trend	*p* between ICT and CSII 2019
ICT	85 (78.7)	70 (59.3)	72 (59.0)	58 (66.7)	285 (65.5)	0.063	
CSII	17 (15.7)	39 (33.1) *	37 (30.3)	28 (32.2)	121 (27.8)	0.004	
HbA1c (%)							0.399
ICT	8.0 (1.9)	7.8 (1.5)	7.4 (1.4)	8.0 (1.8)	7.7 (1.6)	0.673	
CSII	6.9 (1.1)	7.7 (1.3)	6.9 (1.2)	7.6 (1.5)	7.3 (1.3)	0.771	
Non-severe hypoglycaemia (events/week)							0.507
ICT	n.a.	2.25 (2.88)	2.35 (2.66)	2.12 (3.12)	2.24 (2.84)	0.550	
CSII	n.a.	3.71 (1.34)	3.09 (2.43)	2.59 (2.65)	3.06 (2.40)	0.089	
Severe hypoglycaemia (events/year)							0.133
ICT	0.14 (0.54)	0.18 (0.69)	0.22 (0.76)	0.44 (1.12)	0.23 (0.80)	0.581	
CSII	0.06 (0.25)	0.18 (0.82)	0.16 (0.5)	0.15 (0.54)	0.15 (0.61)	0.759	

* = *p* < 0.05 compared to previous survey period.

## Data Availability

The data presented in this study are available upon request from the corresponding author. The data are not publicly available due to privacy.

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
