# Peer review of "Metabolic Control, Diabetic Complications and Drug Therapy in a Cohort of Patients with Type 1 and Type 2 Diabetes in Secondary and Tertiary Care between 2004 and 2019"

_ijerph, 2023, doi:10.3390/ijerph20032631_

Round 1
Reviewer 1 Report
Firstly, I would like to congratulate the authors for collection and execution of a large amount of data meticulously. This manuscript provides a perfect evidence for improved diabetic health care over the period of time in Germany. Being said that I have couple of observations which might improve the quality of the paper.
1. one interesting observation from this data is the average age of T1D patients is 40+, I am curious to know if data sample also include young adults and patients with juvenile diabetes, especially in T1D?
2. According to data presented, diabetic neuropathy is higher in TC cases treated with insulin. Inclusion of a brief background about insulin induced nephro/neuropathy for the benefit of readers.
3. In table.1, space needed before subheading “type2diabetes”.
4. According the manuscript, hypoglycemic conditioned are reported to be higher in T1D, is it related to insulin overdose? Please explain
Reviewer 2 Report
The aim of the study entitled “Metabolic Control, Diabetic Complications and Drug Therapy 2 in a Cohort of Patients with Type 1 and Type 2 Diabetes in Sec- 3 ondary and Tertiary Care between 2004 and 2019” was to obtain a more detailed overview regarding the time-related development of patient characteristics, complications and drug utilization among patients with T1DM and T2DM in secondary care and tertiary care to observe and analyze trends and developments over a period of 15 years.
The present study has the form of a retrospective cross-sectional analysis. The title and abstract are appropriate for the content of the text. Overall, the article is well-constructed. The manuscript is well-presented.
Unfortunately, there are some fundamental concerns with the experimental design. I deem it unlikely that diabetic complications trends are assessed appropriately in the manuscript, if the data for statistical analysis were obtained only from digital medical records.
The Authors reported there were no significant changes in the prevalence of retinopathy over a period of 15 years in both T1DM and T2DM patients. In turn, the prevalence of other complications increased significantly (for example in secondary care for nephropathy in both T1DM and T2DM patients, neuropathy in both T1DM and T2DM patients, and cardiovascular events in T1DM patients).
The Authors admitted that the prevalence of nephropathy may have been overestimated due to the retrospective design, nephropathy could have had a differing etiology from diabetes, and renal function can be influenced by other factors, including the state of hydration or an usage of nephrotoxic drugs. In my opinion, the prevalence of other diabetic complications, including neuropathy or cardiovascular complications may be also inappropriately estimated due to similar bias factors. It is well known that there is a significant association between diabetic complications including retinopathy, peripheral neuropathy, and nephropathy, as well as their severities. Retinopathy is the most common micro vascular complication, followed by neuropathy. There is a significant correlation between retinopathy, nephropathy and neuropathy in association with the duration and control of blood glucose level. It has been documented that the association of severity of diabetic retinopathy with severity of diabetic nephropathy and diabetic neuropathy can be used as a marker for future chronic kidney diseases progression and also to prognosticate neurological outcomes in diabetic patients. Therefore, if the Authors observe the lack of significant changes in the prevalence of retinopathy together with the lack of the changes in HbA1c levels over the period of 15 years, the increasing prevalence of other complications is highly doubtful. This may mean the conclusions put forward by this manuscript are not warranted and I cannot approve the manuscript in this form.
Reviewer 3 Report
The paper entitled “Metabolic Control, Diabetic Complications and Drug Therapy in a Cohort of Patients with Type 1 and Type 2 Diabetes in Secondary and Tertiary Care between 2004 and 2019”, includes potentially relevant data for the clinical aspects of pharmacology directed to medical family care, and especially for dialectological care. The Authors of this publication conclude among others that: ”…. Low rates of hypoglycaemia, lower limb amputations and good glycaemic control in secondary care patients indicate a good structure of patient care…”.
Remarks:
1. I suggest to prepare the Introduction section upon the current/up-to-date literature data (i.e.
3. Action to Control Cardiovascular Risk in Diabetes Study Group; Gerstein, H.C.; Miller, M.E.; Byington, R.P.; Goff, D.C., Jr.; Bigger, J.T.; Buse, J.B.; Cushman, W.C.; Genuth, S.; Ismail-Beigi, F., et al. Effects of intensive glucose lowering in type 2 diabetes. N Engl J Med 2008, 358, 2545-2559, doi:10.1056/NEJMoa0802743.
4. Inzucchi, S.E.; Bergenstal, R.M.; Buse, J.B.; Diamant, M.; Ferrannini, E.; Nauck, M.; Peters, A.L.; Tsapas, A.; Wender, R.; Matthews, D.R., et al. Management of hyperglycemia in type 2 diabetes: a patient-centered approach: position statement of the American Diabetes Association (ADA) and the European Association for the Study of Diabetes (EASD). Diabetes Care 2012, 35, 1364-1379, doi:10.2337/dc12-0413.
5. Bundesärztekammer (BÄ K); Kassenärztliche Bundesvereinigung (KBV); Arbeitsgemeinschaft der Wissenschaftlichen Medizinischen Fachgesellschaften (AWMF). Nationale VersorgungsLeitlinie Therapie des Typ-2-Diabetes – Langfassung, 1. Auflage. Version 4. 2013, zuletzt geändert: November 2014. Availabe online: www.dm-therapie.versorgungsleitlinien.de (accessed on 05.01.2022).)
2. The Conclusions section does not answer the task/questions indicated in the aim of the study – for example, the issue of drug utilization was not addressed. Conclusions and aim must be compatible.
3. Sentences are imprecise – “…Statistical analysis was performed using a linear mixed model for continuous variables. A generalized linear mixed model was 113 used for categorical variables…” (lines 112-114) – should be clarified.
4. The Authors need to indicate/clarify which active pharmaceutical substances from the “Antihypertensive drugs” group were taken by patients.
5. The Authors need to indicate/clarify which active pharmaceutical substances from the “Antihyperlipidaemic drugs” group were taken by patients.
6. The authors omit the parameters of lipid metabolism – Total cholesterol, triglycerides, LDL-cholesterol, HDL-cholesterol – absolutely must be supplemented.
7. The Authors omit the parameters of inflammation assessment – crucial in the course of type 2 diabetes mellitus – it needs to be supplemented.
Round 2
Reviewer 2 Report
Dear Authors,
Thank you for your comments.
Congratulations.
Reviewer 3 Report
The Authors of the manuscript entitled "Metabolic Control, Diabetic Complications and Drug Therapy in a Cohort of Patients with Type 1 and Type 2 Diabetes in Secondary and Tertiary Care between 2004 and 2019" optimally modified the article submitted to the editorial office of the International Journal of Environmental Research and Public Health.
The article in its current form is suitable for publication without changes.